# Enhancement of Mechanical and Bond Properties of Epoxy Adhesives Modified by SiO_2_ Nanoparticles with Active Groups

**DOI:** 10.3390/polym14102052

**Published:** 2022-05-18

**Authors:** Jiejie Long, Chuanxi Li, You Li

**Affiliations:** 1School of Civil Engineering and Architecture, Changsha University of Science and Technology, No. 960 Wanjiali Road, Changsha 410114, China; longjiejie5391@163.com; 2School of Civil Engineering, Hunan University of Technology, Zhuzhou 412007, China

**Keywords:** epoxy adhesives, mechanical properties, SiO_2_ nanoparticles, active groups, CFRP/steel double lap joints

## Abstract

In order to improve the mechanical and bond properties of epoxy adhesives for their wide scope of applications, modified epoxy adhesives were produced in this study with SiO_2_ nanoparticles of 20 nm in size, including inactive groups, NH_2_ active groups, and C_4_H_8_ active groups. The mechanical properties of specimens were examined, and an investigation was conducted into the effects of epoxy adhesive modified by three kinds of SiO_2_ nanoparticles on the bond properties of carbon fiber reinforced polymer and steel (CFRP/steel) double lap joints. According to scanning electron microscopy (SEM), the distribution effect in epoxy adhesive of SiO_2_ nanoparticles modified by active groups was better than that of inactive groups. When the mass fraction of SiO_2_-C_4_H_8_ nanoparticles was 0.05%, the tensile strength, tensile modulus, elongation at break, bending strength, flexural modulus, and impact strength of the epoxy adhesives reached their maximum, which were 47.63%, 44.81%, 57.31%, 62.17%, 33.72%, 78.89%, and 68.86% higher than that of the EP, respectively, and 8.45%, 9.52%, 9.24%, 20.22%, 17.76%, 20.18%, and 12.65% higher than that of the inactive groups of SiO_2_ nanoparticles, respectively. The SiO_2_ nanoparticles modified with NH_2_ or C_4_H_8_ active groups were effective in improving the ultimate load-bearing capacity and bond properties of epoxy adhesives glued to CFRP/steel double lap joints, thus increasing the strain and interface shear stress peak value of the CFRP surface.

## 1. Introduction

Due to its excellent mechanical properties, bond properties, insulation properties, thermal stability, alkali resistance, stable chemical properties, and low cost, epoxy adhesive is now widely used in bonding metal or non-metal with thermosetting materials [1,2,3,4,5,6], such as the bonding layer between ultrahigh performance concrete (UHPC) and a thin asphalt layer, or the bond of carbon fiber reinforced polymer (CFRP) with fatigue damaged steel structures. However, such examples require epoxy adhesives to have excellent mechanical properties, surface roughness, and toughness, yet unmodified epoxy adhesive composites are too brittle and vulnerable to cracking [7,8,9]. The aforementioned properties constrain the applications of epoxy adhesives in civil engineering, military industry, national defense, chemical industry, aerospace, and other high-technology industries [10,11]. Therefore, it is necessary to explore how to improve the mechanical properties of epoxy adhesives effectively [12,13,14,15,16,17,18].

A common method of reinforcing and toughening epoxy resin is to introduce a suitable filler. The surface atoms of nanomaterials are highly unsaturated, which makes them capable of reaction with the active groups in the epoxy adhesives to form a force. Therefore, many researchers have shown considerable interest in determining the optimal nanofiller. In [19], the authors studied the effects of adding a mineral filler on the mechanical properties of epoxy composites, and it was proven that the addition of ocher to the epoxy composite provides an increase in the thermal stability of the composite. Fouly et al. [20] found that the fracture toughness and ductility were enhanced by 47.3% and 12%, respectively, for the epoxy composite modified by 0.4 wt% alumina nanoparticles. Interestingly, using a fly ash filler classified as industrial waste, the results with fly ash/epoxy composite showed that the tensile strength increased with fly ash content up to a critical threshold, and then decreased with higher ash content [21]. As a result, the composite material leads to the development of more micro-cracks in different directions, acting as a toughening agent [22,23,24,25,26,27,28,29,30,31,32,33].

However, nanomaterials can cause small size effects and tend to agglomerate when mixed with epoxy resin, thus resulting in a high heterogeneity of composite materials and causing stress concentration in the process of bearing loads. In [34], the field emission scanning electron microscope images confirmed that nanocomposites with organo-modified MMT loading beyond 1 wt% exhibit large tactoid particles and agglomeration. In addition to diminishing the improvement of mechanical properties for composite materials, this also leads to an increased viscosity and difficulties in production [35,36]. The modification of the surface of the nanomaterials by a large number of active groups is capable of improving the compatibility between nanomaterials and epoxy adhesives. The nanomaterial can be effectively dispersed in epoxy adhesives, which is conducive to improving the mechanical properties of the composite [37,38,39,40,41]. In most of the existing research on epoxy adhesive toughening, the focus is placed on adjusting the mix ratio of nanomaterials and functionalizing coupling agents to improve the mechanical properties of epoxy adhesives [42,43,44,45]. Selimov et al. [46] examined the effects of silane coupling agents on the dispersion of particles in hybrid carbon fiber composites, and they found that the surface treatments improved the dispersion of the particles and reduced sedimentation. In [47], the authors reported the effects of silane coupling agent (3-aminopropyl triethylsilane) on alumina particles, which were subsequently used as fillers for an epoxy resin matrix. Results from mechanical testing indicated improvements in terms of mechanical properties as a result of the coupling treatment. Aksimentyeva et al. [48] investigated the surface modification of magnetite-polymer nanoparticles by luminescent BaZrO_3_ nanocrystals and polyaniline as conducting substances, and the proposed method of surface modification may be used for developing sensors and functional materials for diagnostic methods in medicine. Tsebriienko et al. [49] proposed the poly (titanium oxide) of interpenetrating polymer networks, which showed that the presence of poly (titanium oxide) increases the compatibility of the components of the interpenetrating polymer. Another interesting finding reported in [50] is about the comparison of the mechanical properties of untreated and surface-modified alumina/epoxy nanocomposites, which corroborate the sensitivity of the strength-to-particle surface modification.

SiO_2_ nanoparticles are hydrophilic and readily agglomerate in epoxy adhesives due to the fact that the surface of SiO_2_ nanoparticles contains a large amount of hydroxyl (–OH). Therefore, it is necessary to modify the surface of SiO_2_ nanoparticles to enhance their compatibility with epoxy adhesives and improve the effect of SiO_2_ nanoparticles on the mechanical properties of epoxy adhesives. The modification methods of SiO_2_ nanoparticles include the alcohol esterification method, the silane coupling agent method, the polymer graft modification method, the surfactant method, the in situ modification method, and the high-energy modification method, among which the most commonly used and traditional method is the silane coupling agent method. The silane coupling agent is a chemical substance with a bidirectional reaction function that can make the bonding interface of epoxy adhesive and SiO_2_ nanoparticles into a chemical bond, which can significantly improve the reinforcement performance of SiO_2_ nanoparticles. In [51], the study investigated the epoxy-based composites containing SiO_2_ modified by various silane coupling agents, which showed that the mechanical properties of the modified composite were improved by incorporation of the silane coupling agents at lower silica content, especially the failure strain. In [52,53], the studies found that the coupling agents form hydrogen bonds with cellulose chains, which can enhance the interaction energy between nano-SiO_2_ and cellulose chains, increasing the adsorption between nano-SiO_2_ and cellulose chains. In another interesting study, ref. [54] showed that the dichlorodimethylsilane-modified SiO_2_ has a higher content of organic groups and a higher grafting density than the phenyltrimethoxysilane-modified SiO_2_. As suggested by Li et al. [30], the modified SiO_2_ nanoparticles contain a large number of active groups on the surface, which can improve the impact strength and shear yield of the composite by reacting with the epoxy groups in the epoxy adhesive matrix. To sum up, the introduction of active groups in modified SiO_2_ nanoparticles is capable of improving the mechanical properties of epoxy adhesive matrix composites, which provides an effective solution to their reinforcement and toughening.

Allowing for the weakness of the existing research, this paper aims to reveal the impact of several nanomaterials modified by active groups on the mechanical characteristics of adhesives. An investigation was conducted into the mechanical properties of epoxy adhesives as modified by adding three nanoparticles, SiO_2_ nanoparticles, SiO_2_-NH_2_ nanoparticles, and SiO_2_-C_4_H_8_ nanoparticles: the properties investigated include the stress–strain constitutive relationship, tensile strength, elastic modulus, elongation at break, shear strength, impact strength, and so on. In addition to comparing the composition ratio in the epoxy adhesive, ultrasonic cell fragmentation and dispersion technology were applied to further improve the mechanical properties of the nano-modified epoxy adhesive, and scanning electron microscopy (SEM) was used to indicate the wear mechanism. Then, the modified adhesive was applied to a CFRP-reinforced steel plate to reveal the effects of SiO_2_ nanoparticles as modified using the inactive groups, and of SiO_2_ nanoparticles modified by NH_2_ groups and C_4_H_8_ groups, on the interface bond properties of lap joints, such as ultimate load-bearing capacity, interfacial shear stress, and maximum stress transfer range. Finally, the optimum nanoparticles and extent for steel plate reinforcement were determined.

## 2. Materials and Methods

### 2.1. Preparation of Test Materials

Epoxy adhesive (E-51 bisphenol A; specific equivalent weight: 192–196 g/mol; viscosity at 25 °C: 11,000~14,000 mPa.s; density at 25 °C: 1.6–2.3 g/cm^3^; obtained from Changsha Nan Xing Chemical Technology Co., Ltd., of SINOPEC, Changsha, China); 3-glycidoxypropyltrimethoxysilane (KH-560, Changsha Chemical Research Institute, Changsha, China); curing agent (viscosity at 25 °C: 500~1000 mPa.s obtained from Changsha Chemical Research Institute, China); SiO_2_ nanoparticles (Bongrui New Material Technology Co., Ltd., Fuyang, China). Before the experiments in this paper, relevant studies have been carried out on SiO_2_ nanoparticles with different mass fractions to modify epoxy adhesive, and the mass fractions of 0.02%, 0.05%, 0.10%, 0.20%, and 0.50% were selected as the values to be presented. First, 120 g of epoxy adhesive was added to specified amounts of SiO_2_ nanoparticles (0.024 g, 0.06 g, 0.12 g, 0.24 g, 0.60 g) and 3 g of coupling agent. After 10 min of manual dispersion, it was put into an ultrasonic cell crusher (Xinzhi ultrasonic cell crusher was procured from Ningbo Xinzhi Biotechnology Co., Ltd., Ningbo, China) for 2 h. Then, after the mixture was fully dispersed, 35 g of curing agent was added and thoroughly stirred. Next, the mixture was poured into the specimen mold at a uniform speed, and the specimen was placed horizontally indoors and cured at room temperature for 7 days. The production process of the epoxy adhesive specimen is shown in Figure 1, and the detailed parameters of SiO_2_ nanoparticles are displayed in Table 1.

### 2.2. Measurement of Tensile Properties

The tensile properties of epoxy adhesives were tested in accordance with ASTM D638-10 [55], and five specimens were prepared for each ratio. The dimensions of the specimen are shown in Figure 2.

The quasi-static tensile properties of epoxy adhesive were tested using a 50 kN electronic universal testing machine (WDW-300D; Ji’nan Ke Sheng Testing Machine Equipment Co., Ltd., Ji’nan, China). The specimen was clamped between the upper and lower fixtures of the testing machine shown in Figure 3. Slippage between specimen and fixtures was prevented. Meanwhile, the central axis of the specimen was consistent with the central line of the upper and lower fixtures to ensure axis loading. Thereafter, the extensometer was installed in the middle section of the specimen. The test was uniformly and continuously loaded at a rate of 2 mm/min until failure. At that time, the failure load value was recorded, and the thickness at the middle and at both ends of the tensile section was measured and averaged.

### 2.3. Measurement of Flexural Properties

The flexural properties of epoxy adhesives were tested in accordance with ASTM D790-10 [56], and five flexural specimens with dimensions of 10 mm × 5 mm × 120 mm were made for each ratio. The diagram of the specimen is shown in Figure 4.

The flexural properties of epoxy adhesives were tested using a 50 kN electronic universal testing machine. A three-point bending test was used for the loading shown in Figure 5. The three loading points of the specimen were 10 mm from the left edge, the center point, and 10 mm from the right edge, respectively. The loading rate and data processing were consistent with the tensile test.

### 2.4. Measurement of Impact Performance

The impact properties of epoxy adhesives were tested in accordance with GB/T 1043.1-2008 [57], and five impact specimens with type A notch were made for each ratio. A diagram of one such specimen is shown in Figure 6.

The impact performance of epoxy adhesives was tested using a liquid crystal plastic pendulum impact testing machine (LZ21.400-B; Shenzhen Lambo Sansa Material Testing Co., Ltd., Shenzhen, China). A simple beam bearing with a span of 60 mm was selected in the test. The pendulum impact blade was exactly in the middle of the span. When loaded, the specimen extended beyond the left and right supports by 10 mm. The pendulum was turned counterclockwise by hand to the end and then released quickly, and the specimen was struck clockwise by the pendulum.

### 2.5. Measurement of Shear Performance

The shear properties of epoxy adhesives were examined according to ISO 4587-2003 international standards [58], and five specimens were prepared for each ratio. The dimensions of the specimen are shown in Figure 7.

The shear properties of epoxy adhesives were tested using a 50 kN electronic universal testing machine. To verify that no torque was generated, a piece of equal-thickness steel plate was clamped at both ends. The test was consistently and continuously loaded at a rate of 1 mm/min until failure, and the failure load value was recorded at that time. The loading device is shown in Figure 8.

### 2.6. Measurement of Bond Property

In terms of form and dimension, the design of the specimens for the tensile test of CFRP/steel double lap adhesive joints is referred to as type A in ASTM D3528-96 (R2008) [59]. The form of specimens and the installation position of sensors are shown in Figure 9. The total length of the specimens was 500 mm, the length of the steel plate used as bonding was 350 mm, and the length of the small steel block used as loading clamping was 70 mm. Three specimens were prepared for each ratio.

The tensile test of CFRP/steel double lap joints was carried out on a 300 kN quasi-static tensile test machine with displacement control for loading at a rate of 0.3 mm/min. The loading device is shown in Figure 10. The stress and strain data were collected by a static strain testing system.

## 3. Results and Discussion

### 3.1. Analysis of Mechanical Properties of Epoxy Adhesives Modified by SiO_2_ Nanoparticles with Active Groups

#### 3.1.1. Tensile Property

Figure 10 shows the relationship between the amounts of SiO_2_ nanoparticles with different active groups and the tensile mechanical properties of epoxy adhesives. Figure 11a shows that the tensile strength of the epoxy adhesives were modified by SiO_2_-0 nanoparticles, SiO_2_-NH_2_ nanoparticles and SiO_2_-C_4_H_8_ nanoparticles. The improvement in tensile strength of epoxy adhesives was greatest at the same mass fraction (0.05%) of SiO_2_-NH_2_ nanoparticles, SiO_2_-C_4_H_8_ nanoparticles, and SiO_2_-0 nanoparticles, which was 59.04 MPa, 59.05 MPa, and 54.45 MPa, respectively. The former two improved by 47.60% and 47.63% when compared to the EP, and improved by 8.43% and 8.45% when compared to SiO_2_-0 nanoparticles. The tensile modulus of the epoxy adhesives modified with SiO_2_-NH_2_ nanoparticles or SiO_2_-C_4_H_8_ nanoparticles was greater than that of SiO_2_-0 nanoparticles, as shown in Figure 11b. When 0.05% SiO_2_-NH_2_ nanoparticles or SiO_2_-C_4_H_8_ nanoparticles were added into the epoxy adhesives, the tensile modulus of the epoxy adhesives reached a peak value of 6438 MPa or 6517 MPa, respectively, which was improved by 43.06% and 44.81% when compared to the EP, and improved by 8.20% and 9.52%, respectively, when compared to SiO_2_-0 nanoparticles. The elongation break curves (Figure 11c) were similar to those showing the specimens’ tensile strength, with an increased trend in the early stages and a progressive drop in the late stages. The elongation at break of epoxy adhesives modified by SiO_2_-NH_2_ nanoparticles, SiO_2_-C_4_H_8_ nanoparticles or SiO_2_-0 nanoparticles reached its peak value at the same mass fraction (0.05%), which was 2.66%, 2.72% or 2.49%, respectively. The former two were 53.52% and 57.31% higher compared to the EP, and 6.83% and 9.24% higher compared to SiO_2_-0 nanoparticles, respectively. With the increase in nanoparticles, the elongation at break of SiO_2_-C_4_H_8_ decreased very slowly and remained above 2.20%. The addition of small amounts of nanoparticles could greatly improve the tensile properties of epoxy adhesive. With the increase in nanoparticles, the tensile properties of epoxy adhesive tended to increase at first and then decrease. It is worth noting that the tensile strength, the tensile modulus, and the elongation at break of composites were lower than those of EP when the mass fraction of SiO_2_-0 increased to 0.5%. The excessive addition of SiO_2_ nanoparticles was difficult to disperse effectively in the epoxy adhesive and prone to agglomeration, which resulted in stress concentration and a sharp decline in the tensile properties of the composites. The tensile properties of epoxy adhesives modified with SiO_2_-C_4_H_8_ nanoparticles or SiO_2_-NH_2_ nanoparticles were greater than those of SiO_2_-0 nanoparticles, a result which indicates that the nanoparticles modified by active groups have better compatibility with epoxy adhesives than those modified by inactive groups.

Figure 11d depicts the stress–strain curves of tensile tests of the EP and three types of SiO_2_ nanoparticles with a mass of 0.05%. At the initial stage of loading, the nanoparticles were closely bonded to the resin matrix, no damage occurred in the epoxy adhesives, and the strain–stress showed a linear relationship. As the stress continued to increase, the integrity of the adhesives was gradually destroyed, the strain increment became larger and larger, and the slope decreased gradually. From the graph, it could be seen that the failure stress and slope of the colloidal nanoparticle epoxy adhesives were clearly higher than those of the EP. Meanwhile, SiO_2_ nanoparticles containing active groups had a better effect on the tensile properties of the epoxy adhesives than those of SiO_2_ nanoparticles containing inactive groups. Therefore, SiO_2_ nanoparticles modified by NH_2_ and C_4_H_8_ active groups had little difference in improving the tensile properties of epoxy adhesives, and SiO_2_-C_4_H_8_ nanoparticles were slightly better than SiO_2_-NH_2_ nanoparticles in improving the tensile properties of the epoxy adhesives.

The SEM diagram of the tensile specimen of pure epoxy adhesive was smooth and the crack development was single, as shown in Figure 12a. Figure 12b–d shows that the cross-section of epoxy adhesive with a mass fraction of 0.05% SiO_2_ nanoparticles was much coarser than that of EP, and there were obvious folds. The cross-section in Figure 12b contained some aggregation, which indicates that the compatibility of nanoparticles modified by active groups with epoxy resin was better than that of nanoparticles with inactive groups. The SiO_2_ nanoparticles in the colloidal section in Figure 12c,d were evenly dispersed, with irregular and disordered fracture sections and a large number of microcracks, which is typical of ductile fracture morphology. The cross-section of Figure 12e was smoother than that of Figure 12b–d, with a large convex-shaped area on the surface, and the crack development tended to be single. The reason for this is that SiO_2_ nanoparticles were not evenly dispersed in the epoxy adhesive and agglomeration occurred, which reduced the colloidal mechanical properties.

#### 3.1.2. Flexural Properties

Figure 13 shows the relationship between the amounts of SiO_2_ nanoparticles with different active groups and the flexural properties of epoxy adhesives. Figure 13a demonstrates that the bending strength of epoxy adhesives modified with SiO_2_ nanoparticles was greater than that of the EP. The bending strength of the epoxy adhesives reached a peak value of 79.40 MPa or 86.71 MPa when 0.05% SiO_2_-NH_2_ nanoparticles or SiO_2_-C_4_H_8_ nanoparticles were added into the epoxy adhesives, respectively. They were 48.50% and 62.17% higher compared to the EP, and 10.08% and 20.22% higher compared to SiO_2_-0 nanoparticles, respectively. The flexural modulus of the epoxy adhesives modified by SiO_2_-NH_2_ nanoparticles or SiO_2_-C_4_H_8_ nanoparticles was superior to that of SiO_2_-0 nanoparticles as shown in Figure 13b. The improvement of epoxy adhesives modified by SiO_2_-NH_2_ nanoparticles, SiO_2_-C_4_H_8_ nanoparticles, and SiO_2_-0 nanoparticles reached its peak value at the same mass fraction (0.05%), which was 4620.97 MPa, 5019.86 MPa, and 4262.88 MPa respectively. Compared with the EP, these improvements were by 23.09%, 33.72%, and 13.55%, respectively.

In general, the addition of a small amount of SiO_2_ nanoparticles can greatly improve the bending properties of epoxy adhesives. The bending properties of epoxy adhesives modified by SiO_2_ nanoparticles with active groups were better than those with inactive groups. When the mass fraction of SiO_2_-C_4_H_8_ nanoparticles was 0.05%, the bending property of epoxy adhesive reached its best effect. When the mass fraction of SiO_2_ nanoparticles exceeded 0.05%, the bending properties of the epoxy adhesives modified bySiO_2_-NH_2_ nanoparticles decreased faster than those of SiO_2_-C_4_H_8_ nanoparticles. The flexural properties of the composite were lower than those of EP when the mass fraction of SiO_2_-NH_2_ nanoparticles was 0.50%, which also indicates that the distribution effect of the C_4_H_8_ active group in epoxy adhesive was better than that of the NH_2_ active group.

#### 3.1.3. Impact Performance

At the moment of contact between the impact specimen and the pendulum, the pendulum immediately cut the specimen into two sections along the V-notch, accompanied by a few fragments of sputtering, and the crack extended perpendicularly to the direction of load. As shown in Table 2, the impact strength of the epoxy adhesives modified by SiO_2_ nanoparticles under different mass fractions was analyzed. When 0.02% SiO_2_ nanoparticles were added to epoxy adhesives, the impact strength of the epoxy adhesives was significantly improved. With the increase in nanoparticles, the impact strength of epoxy adhesives steadily increased, and reached its peak value when the mass fraction of SiO_2_ nanoparticles was 0.05%. The epoxy adhesive modified by 0.05% SiO_2_-C_4_H_8_ nanoparticles had the best effect on improving the impact strength (32.76 kJ/m^2^), which was 78.89% higher compared to the EP (18.31 kJ/m^2^) and 20.18% higher compared to the SiO_2_-0 nanoparticles (27.26 kJ/m^2^). When the mass fraction of SiO_2_ nanoparticles was 0.10%, the impact properties of the three kinds of epoxy adhesives modified by SiO_2_ nanoparticles all decreased, which might have been due to the large amounts of SiO_2_ nanoparticles that were difficult to disperse effectively in the epoxy adhesive and prone to agglomeration.

#### 3.1.4. Shear Performance

Figure 14 shows the load-displacement curves of shear specimens modified by 0.05% SiO_2_-C_4_H_8_ nanoparticles, 0.05% SiO_2_-NH_2_ nanoparticles, 0.05% SiO_2_-0 nanoparticles, and the EP. As can be seen in the figure, the load of the specimens grew quickly as displacement increased at the beginning of the test. When the load value reached about 0.3 kN, the load increment tended to zero with increase in displacement, and the curve became flatter. Thereafter, the load increment was linearly correlated with the displacement increment, and the slope was slightly less than at the beginning of the test. The shear strength of the adhesives was greatly improved after being modified by nanoparticles, and the shear strength of epoxy adhesives modified by SiO_2_-C_4_H_8_ nanoparticles or SiO_2_-NH_2_ nanoparticles was 68.86% (3.52 kN) or 67.93% (3.50 kN) higher compared to the EP (2.09 kN), and 12.65% or 2.03% higher compared to the SiO_2_-0 nanoparticles (3.13 kN), respectively. The shear properties of epoxy adhesive modified by SiO_2_ nanoparticles with the two kinds of active groups were basically the same, and both better than those with no active groups. However, the flatter part of the curve for the C_4_H_8_ active group was significantly longer than that of the NH_2_ active group, which might have been caused by the relative slip between the sheared specimen and the clamp.

### 3.2. Analysis of Bond Properties of Epoxy Adhesives Modified by SiO_2_ Nanoparticles with Active Groups

#### 3.2.1. Load-Displacement Relationship

Figure 15 shows the load-displacement curves of CFRP/steel double lap specimens in the tensile process. With the increase in displacement, the load increment of the three specimens increased. The shear strength and ductility of the epoxy adhesives were improved to a certain extent after the addition of SiO_2_ nanoparticles. It can be seen that there is an abrupt drop in the curves for SiO_2_-0 nanoparticles and EP, which was due to relative slip between the loading device and the specimen during the loading process. As the displacement continued to increase, the specimen was re-loaded. The load peak value of the specimen modified by SiO_2_-C_4_H_8_ nanoparticles, SiO_2_-NH_2_ nanoparticles, and SiO_2_-0 nanoparticles was 57.57 kN, 55.55 kN, and 52.09 kN, which was 41.24%, 36.29% and 27.80% higher compared to the EP (40.76 kN), respectively. The bond properties of the epoxy adhesives modified by SiO_2_ nanoparticles with the two kinds of active groups were better than those with no active groups, and SiO_2_ nanoparticles modified by the C_4_H_8_ active group showed the best improvement.

#### 3.2.2. Surface Strain Distribution of CFRP Plate

The strain distribution on the CFRP surface is shown in Figure 16. Under ultimate load, a considerable stress gradient of the double lap joints bonded by the EP arose near the free end of the steel plate. The strain of EP at 5 mm from the free end of the steel plate was 1339 με; the strain at 15 mm from the free end of the steel plate dropped to 740 με, decreasing by 44.73%; and the strain at 25 mm from the free end of the steel plate was 572 με, decreasing by 57.28%. When SiO_2_-0 nanoparticles with a mass fraction of 0.05% were added into the epoxy adhesives, the strain at 5 mm from the free end of the steel plate was 2750 με; the strain at 15 mm from the free end of the steel plate was 2001 με, decreasing by 27.24%; and the strain at 25 mm from the free end of the steel plate was 1863 με, decreasing by 32.25%. When SiO_2_-NH_2_ nanoparticles with a mass fraction of 0.05% were added into the epoxy adhesives, the strain at 5 mm from the free end of the steel plate was 3113 με; the strain at 15 mm from the free end of the steel plate was 2617 με, decreasing by 15.93%; and the strain at 25 mm from the free end of the steel plate was 2246 με, decreasing by 27.85%. Similarly, the strain at 5 mm from the free end of the steel plate of the epoxy adhesives modified by 0.05% SiO_2_-C_4_H_8_ nanoparticles was 3230 με; the strain at 15 mm from the free end of the steel plate was 2519 με, decreasing by 22.01%. It is worth noting that the maximum strain at 25 mm from the steel plate’s free end was 3002 με, which was considerably higher than the maximum strain at 15 mm, and a decrease of 7.06% compared to that at 5 mm from the free end of the steel plate. The results indicated that SiO_2_ nanoparticles could improve the bond properties of epoxy adhesives, increase the strain and strain gradient range of the CFRP surface, and improve the final bearing capacity of double lap joints. The effect of SiO_2_-C_4_H_8_ nanoparticles and SiO_2_-NH_2_ nanoparticles resulted in a more effective improvement than that of SiO_2_-0 nanoparticles. A tearing sound was noted, which was emitted during the loading procedure of the specimen by the peeling of a small section of the interface. The stripping of the CFRP plate and steel plate occurred as the load increased. A warning signal appeared before the failure of the specimen, which was characteristic of ductile failure.

## 4. Conclusions

This paper proposes epoxy adhesives modified by different active groups of SiO_2_ nanoparticles. With the introduction of SiO_2_-0, SiO_2_-NH_2_, and SiO_2_-C_4_H_8_ nanoparticles into epoxy adhesives at a mass fraction of 0.05%, the improvement in mechanical properties reached its maximum. When added to modify the epoxy adhesives, the SiO_2_-C_4_H_8_ nanoparticles showed the most significant improvement in mechanical properties, while the SiO_2_-0 nanoparticles showed the least improvement. When the mass fraction of SiO_2_-C_4_H_8_ nanoparticles was 0.05%, the tensile strength, tensile modulus, elongation at break, bending strength, flexural modulus, and impact strength were 47.63% (59.05 MPa), 44.81% (6517 MPa), 57.31% (2.72%), 62.17% (86.71 MPa), 33.72% (5019.86 MPa), 78.89% (32.76 kJ/m^2^), and 68.86% (3.52 kN) higher than that of the EP, respectively. SiO_2_ nanoparticles were capable of improving not only the mechanical properties of the epoxy adhesive, but also the strain on CFRP surfaces, the interface shear stress peak, and the transfer range. In general, the mechanical properties and bond properties of epoxy adhesives modified by adding active groups were superior to those modified by adding inactive groups, and the modification effect of C_4_H_8_ active groups was better compared to NH_2_ active groups.

The experiments show that active groups can significantly improve the modification effect of nanoparticles, and that active groups could become a meaningful direction of nanoparticle research for practical engineering applications. The characterization of SiO_2_-NH_2_ and SiO_2_-C_4_H_8_ (FTIR, XRF, etc.) were further studied in the follow-up study.

## Figures and Tables

**Figure 1 polymers-14-02052-f001:**
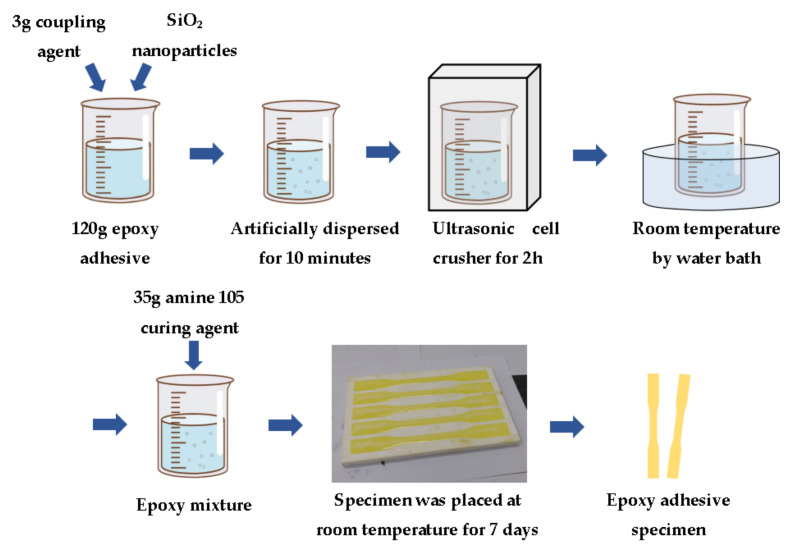
Production process of epoxy adhesive specimen.

**Figure 2 polymers-14-02052-f002:**
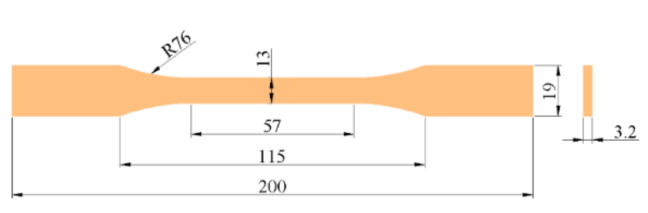
Dimensions of epoxy adhesive tensile specimen (unit: mm).

**Figure 3 polymers-14-02052-f003:**
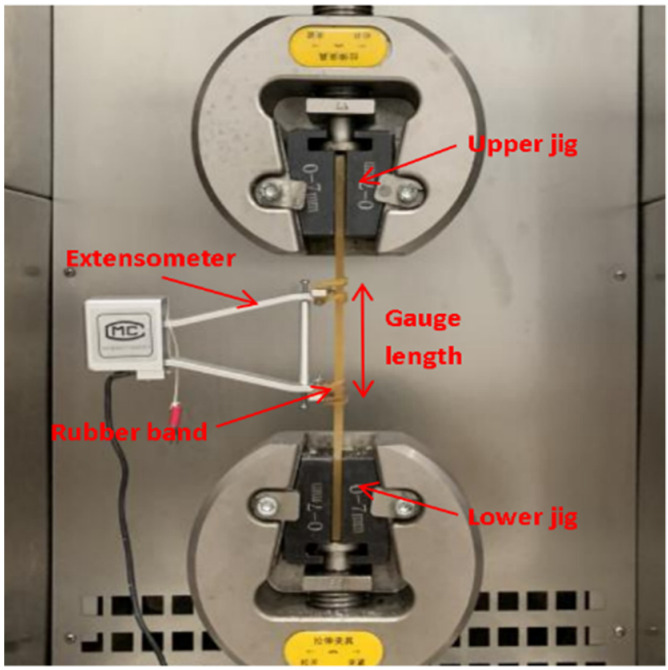
Tensile test loading diagram.

**Figure 4 polymers-14-02052-f004:**
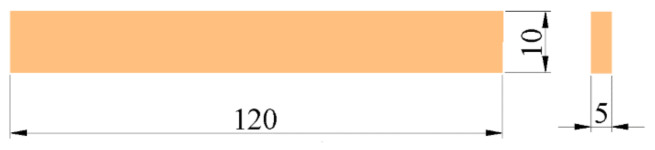
Dimensions of epoxy adhesive flexural specimen (unit: mm).

**Figure 5 polymers-14-02052-f005:**
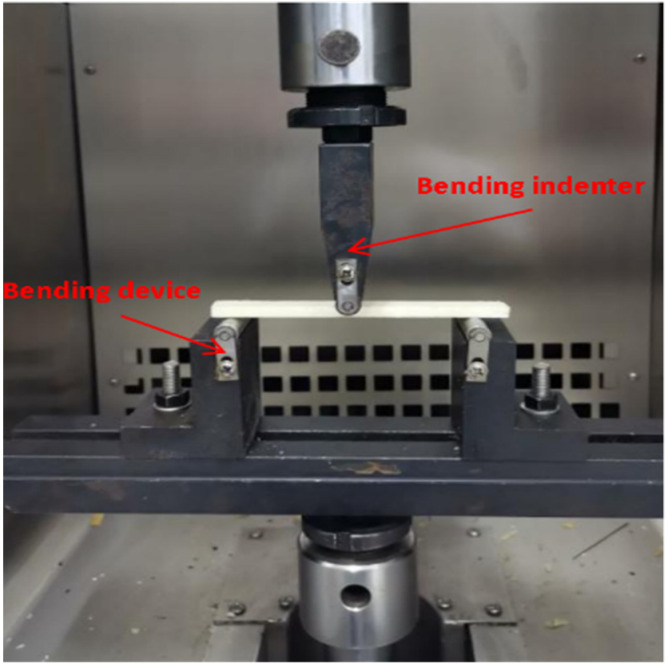
Flexural test loading diagram.

**Figure 6 polymers-14-02052-f006:**
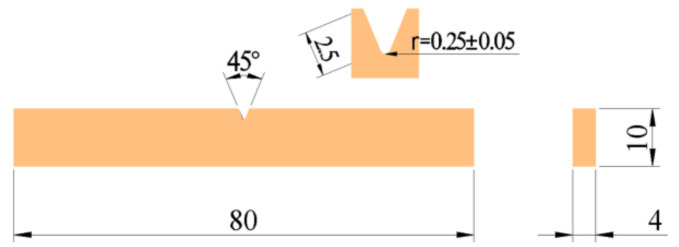
Dimensions of epoxy adhesive impact specimen (unit: mm).

**Figure 7 polymers-14-02052-f007:**
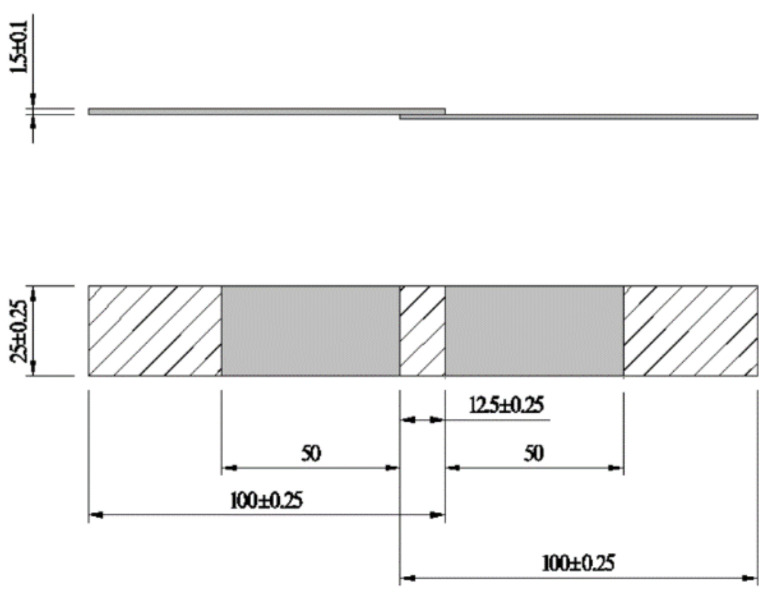
Dimensions of epoxy adhesive shear specimen (unit: mm).

**Figure 8 polymers-14-02052-f008:**
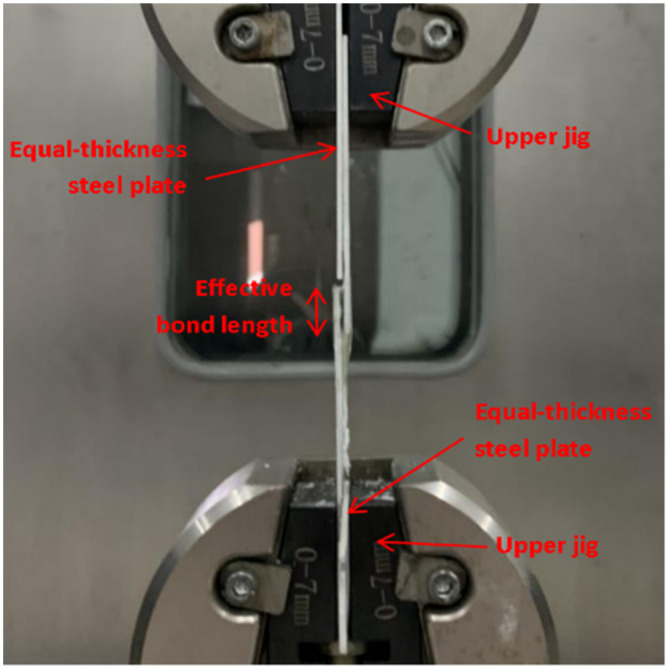
Shear test loading diagram.

**Figure 9 polymers-14-02052-f009:**
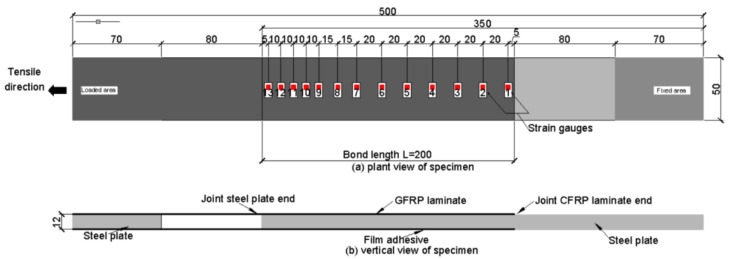
CFRP/steel double lap joints form and strain gauge arrangement (unit: mm).

**Figure 10 polymers-14-02052-f010:**
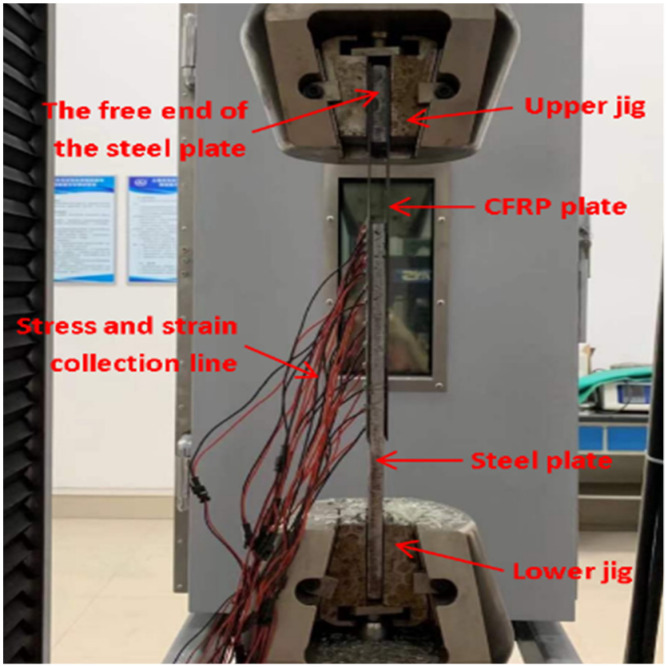
Tensile test of CFRP/steel double lap test piece.

**Figure 11 polymers-14-02052-f011:**
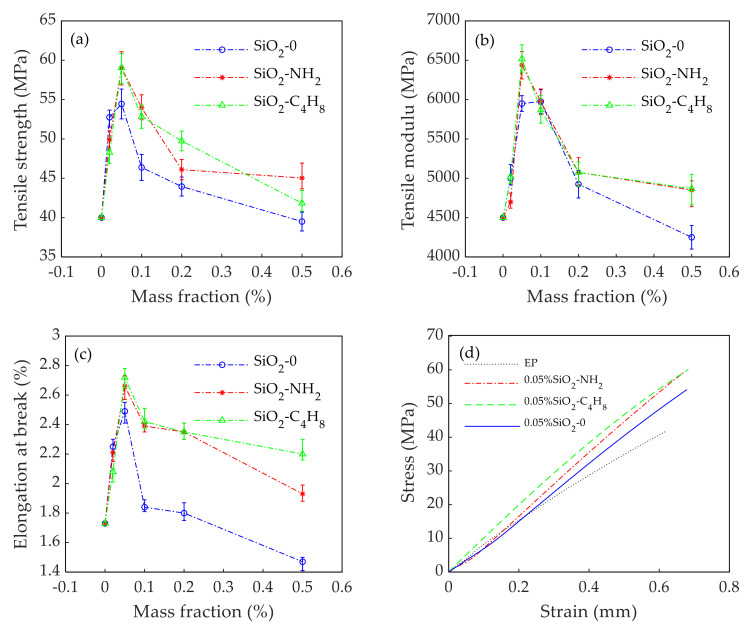
Relationship between the dosages of SiO_2_ nanoparticles with different active groups and tensile mechanical properties of epoxy adhesives: (**a**) tensile strength; (**b**) tensile modulus; (**c**) elongation at break; (**d**) relationship between stress and strain. (The margin of error is within 5%).

**Figure 12 polymers-14-02052-f012:**
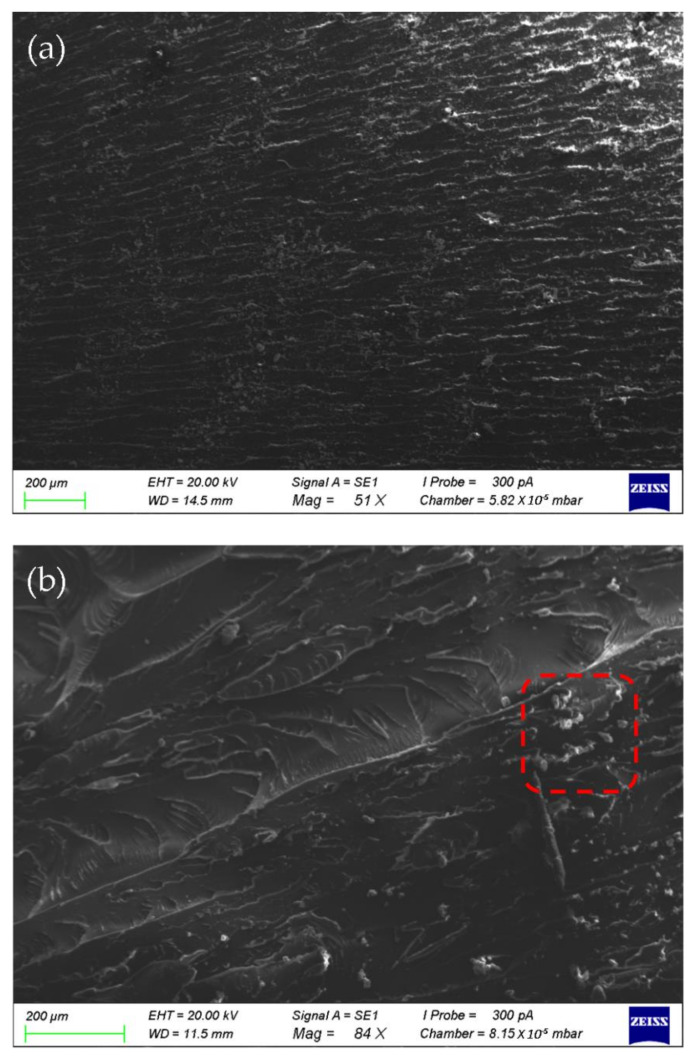
SEM diagram of the section of the adhesive colloid tensile specimens: (**a**) EP; (**b**) 0.05% SiO_2_-0; (**c**) 0.05% SiO_2_-NH_2_; (**d**) 0.05% SiO_2_-C_4_H_8_; (**e**) 0.50% SiO_2_-0.

**Figure 13 polymers-14-02052-f013:**
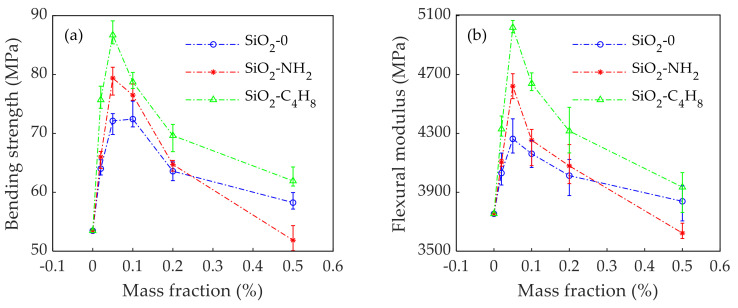
Relationship between the dosages of SiO_2_ nanoparticles with different active groups and bending mechanical properties of epoxy adhesives: (**a**) bending strength; (**b**) flexural modulus. (The margin of error is within 5%).

**Figure 14 polymers-14-02052-f014:**
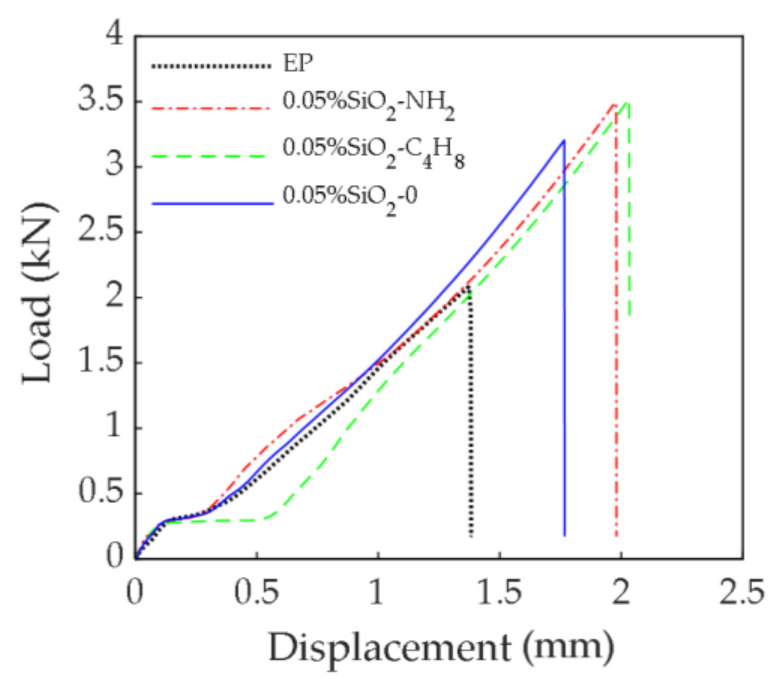
Load-displacement curves of epoxy adhesives modified by nanoparticles with different active groups during shear test.

**Figure 15 polymers-14-02052-f015:**
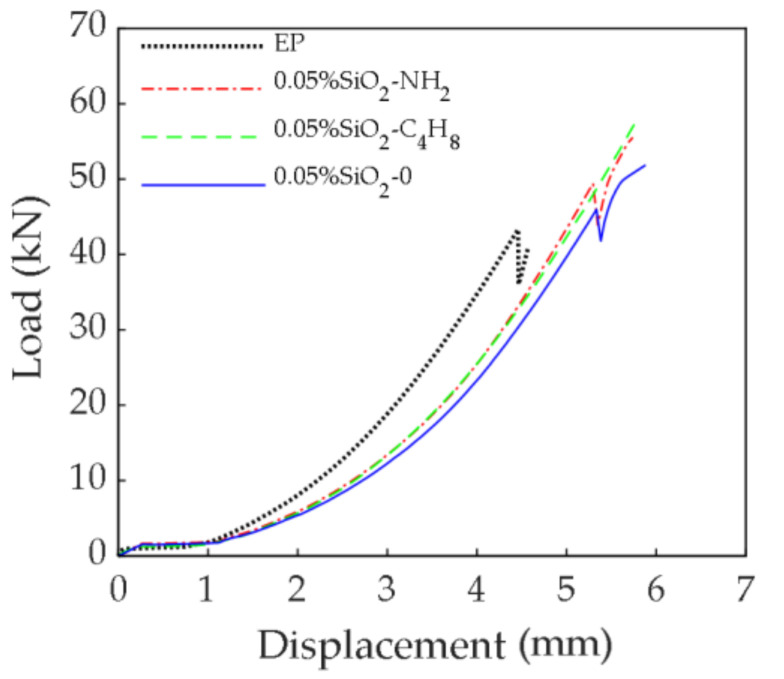
Load-displacement curves of CFRP/steel double lap specimens.

**Figure 16 polymers-14-02052-f016:**
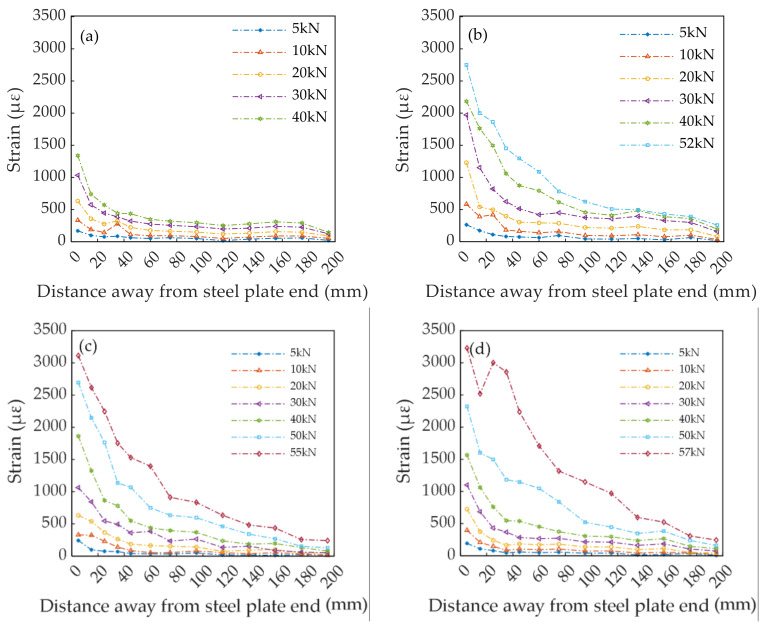
Strain distribution of CFRP surface: (**a**) EP; (**b**) SiO_2_-0 nanoparticles; (**c**) SiO_2_-NH_2_; (**d**) SiO_2_-C_4_H_8_ nanoparticles.

**Table 1 polymers-14-02052-t001:** Detailed parameters of SiO_2_ nanoparticles.

Types	Particle Size	Specific Surface Area	Volume Density	Density	Active Groups
SiO_2_-0	20 nm	230 m^2^/g	0.06 g/cm^3^	2.2–2.6 g/cm^3^	/
SiO_2_-NH_2_					–NH_2_
SiO_2_-C_4_H_8_					–C_4_H_8_

**Table 2 polymers-14-02052-t002:** Impact test data of epoxy adhesives modified by SiO_2_ nanoparticles with different active groups.

Mass Fractions (%)	Impact Strength (kJ/m^2^)		
	SiO_2_-NH_2_	SiO_2_-C_4_H_8_	SiO_2_-0
0	18.31	18.31	18.31
0.02	22.21	23.90	22.59
0.05	28.74	32.76	27.26
0.10	25.81	28.27	24.12

## Data Availability

The characterization data are available upon request from the authors.

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
