# Peer review of "Enhancement of Mechanical and Bond Properties of Epoxy Adhesives Modified by SiO2 Nanoparticles with Active Groups"

_polymers, 2022, doi:10.3390/polym14102052_

Round 1

Reviewer 1 Report

This version does not look worthy and cannot be recommended for publication in this form and at least needs major revision.

  1. Abbreviation CFRP needs to be deciphered!
  2. Line 14. Specify nanoparticles of what size.
  3. Before starting the discussion (Line 157) about SiO2, more information about the last level of research of other polymer-oxide composites should be given.

This will arouse more keen interest among readers working in related fields. See, some axamples:  (TiO2) - Tsebriienko, T.; Popov, A.I. Effect of Poly(Titanium Oxide) on the Viscoelastic and Thermophysical Properties of Interpenetrating Polymer Networks. Crystals 202111, 794. https://doi.org/10.3390/cryst11070794

(Fe3O4) - Aksimentyeva, O. I., Savchyn, et al . Modification of polymer-magnetic nanoparticles by luminescent and conducting substances. Molecular Crystals and Liquid Crystals2014, 590(1), 35-42.

  1. There is one absolutely significant shortcoming of the introduction - this is the lack of a reasonable overview of the recent development on SiO2 nanostructures. What methods can be synthesized, which one is best suited for this work, methods for their characterization? This will also allow a more adequate understanding of the relevance of the work.
  2. Line 169. In this paragraph, it is necessary to justify the accuracy of measurements in hundredths of a percent. The same for Figure 10 (a, b, c).

Reviewer 2 Report

The manuscript under the title: “Enhancement on Mechanical and Bond Properties of Epoxy Adhesives by SiO2 Nanoparticles with Active Groups” is in line with Polymers journal. This topic is relevant and will be of interest to the readers of the journal. It based on original research. This research has scientific novelty and practical significance. The article has a typical organization for research articles.
Before the publication it requires significant improvements, especially:

  1. The "Introduction" section: it has been proven that the effect of fillers on the properties of polymer composites is determined by many factors: ……. I think the related references should be cited corresponding to each aspect, e.g. (but not limited to these), which will undoubtedly improve the "Introduction" section:
  • Polymers 2020, 12(7), 1437; https://doi.org/10.3390/polym12071437
  • International Polymer Science and Technology. 2013, V. 40, N. 7, P. 49-51. https://doi.org/10.1177/0307174X1304000711
  • Polymers 2019, 11, 2012. https://doi.org/10.3390/polym11122012
  • Polymers 2020, 12, 79. https://doi.org/10.3390/polym12010079
  1. Section 2.1. It is necessary to add the physicochemical characteristics of all components - give a table with the main physicochemical and technological properties of epoxy resin, hardener, SiO2
  2. Line 116 - the dimensions do not match the dimensions in Fig.3.
  3. A technique for functionalizing particles SiO2 and obtaining SiO2-NH2 and SiO2-C4H8 should be added to section 2.
  4. Functionalized particles (SO2-NH2 and SiO2-C4H8) should be studied and characterized in detail (SEM, FTIR, XRF, etc.)
  5. In Fig.12-14, it is necessary to add and discuss data for SiO2-NH2.
  6. It is necessary not only to state the increase in the strength properties of epoxy adhesives with the introduction of functionalized particles, but also to try to explain the mechanism of strengthening.
  7. It would be good to study the structure of fractures of epoxy composites, which will allow us to evaluate the effect of particle functionalization on the structure and, accordingly, the properties of composites.

Reviewer 3 Report

The reviewed manuscript investigates the enhancement in the mechanical and bond properties of epoxy adhesives after the addition of various fractions of active groups of SiO2 nanoparticles. The authors modified the SiO2 nanoparticles with two different active groups SiO2-NH2 and SiO2-C4H8 to show the mechanical properties of the adhesives. The authors conducted a series of tests and extracted the stress-strain constitutive relationship, tensile strength, elastic modulus, elongation at break, shear strength, and impact strength. I have found the paper to be interesting. However, some concerns need to be addressed before accepting the paper for publication to improve the readability and clarity of the manuscript:

  • The use of English language is reasonable, however, there are a number of punctuation and grammatical errors; that should be corrected and rephrased using academic English for a better flow of text for reader.
  • Please consider reviewing the abstract and highlighting the novelty. The abstract does not represent all the key information. It should contain answers to some questions, what problem was studied and why is it important? and what conclusions can be drawn from the results? (Please provide specific results and not generic ones). Please use numbers or % terms to clearly shows the results of your experimental work.
  • The introduction is very short and brief. As the author tried to modify the SiO2 by adding active groups, they can cite the following studies;
  1. Wang H, Sun T, Peng C, Wu Z. Effect of different silane coupling agents on cryogenic properties of silica-reinforced epoxy composites. High Perform Polym 2018;30:24–37.
  2. Fouly, Ahmed, and Mohamed G. Alkalla. "Effect of low nanosized alumina loading fraction on the physicomechanical and tribological behavior of epoxy." Tribology International 152 (2020): 106550.
  3. Selimov A, Jahan SA, Barker E, Dackus P, Carolan D, Taylor A, Raghavan S. Silane functionalization effects on dispersion of alumina nanoparticles in hybrid carbon fiber composites. Appl Optic 2018;57:6671–8.
  4. Brown GM, Ellyin F. Mechanical properties and multiscale characterization of nanofiber–alumina/epoxy nanocomposites. J Appl Polym Sci 2011;119:1459–68.
  5. Rashid ESA, Rasyid MFA, Akil HM, Ariffin K, Kooi CC. Effect of (3-aminopropyl) triethylsilane treatment on mechanical and thermal properties of alumina-filled epoxy composites. Proc IME J Mater Des Appl 2011;225:160–9.
  • In the introduction, please consider reporting on studies related to your work from MDPI journals.
  • The preparation of test materials subsection lacks any graphical images or schematics that show the sample preparation technique, real produced samples…etc. This is the section on materials and methods and the authors should provide sufficient graphical information for the readers to better understand their work and what was done in it.
  • In the sample preparation, why did the authors choose those adhesive ratios of 0.02%, 0.05%, 0.10%, 0.20%, and 0.50% to produce their materials? Is it based on other research recommendations or just chosen by the authors for the sake of this experiment?
  • For table 1, I recommend removing the table and inserting the data in the text body.
  • In the measurement of tensile property subsection, the authors mentioned that the tensile mechanical properties of epoxy adhesives were tested in accordance with ASTM D638-10. Please add a reference.
  • In line 99 and 100, the author gave the dimensions of the test specimen. Did the authors select these dimensions according to a standard? Please provide a reference.
  • Line 111, the thickness at any three places in the tensile section was measured. How did you measure it? Please explain.
  • Line 116, five bending specimens with dimensions of 15 mm×5 mm×120 mm. Did the authors select these dimensions according to a standard? Please provide a reference.
  • Line 130, Did the authors select these dimensions according to a standard? Please provide a reference.
  • Line 140, The shear mechanical properties of epoxy adhesives were examined according to ISO 4587-2003 international standards. Please provide a reference.
  • The design of specimens for tensile test of CFRP/steel double lap adhesive joints referred to ASTM D3528-96 (2008). Please provide a reference.
  • It is noticed that 0.05% SiO2-0, SiO2-NH2, and SiO2-C4H8 present a conversion point along with all the mechanical properties. The authors should explain the reasons for such phenomena. It is better to scan the fracture surface using a scanning electron microscope to illustrate the bonds between the epoxy and the additives.
  • It is recommended to perform XRD for the produced specimen to ensure the additives materials with the active groups didn’t chemically react with the epoxy resin. If a chemical reaction occurred peaks of new materials will appear.
  • Conclusions need to be compacted to highlight the outcomings of the scientific paper. Furthermore, the authors didn't mention their future work at the end of the conclusions.

Please, read the text carefully before the next submission of the paper.

Round 2

Reviewer 1 Report

the authors successfully answered all the questions and took into account all the comments, the article can be recommended for publication!

Reviewer 2 Report

The authors considered most of the comments or adequately responded to the remarks contained in the review; therefore, the work may be approved for publication.

Reviewer 3 Report

Many thanks for the revision and for incorporating all suggested changes to the manuscript that are nicely reflected. The authors did a good job to improve the article. I believe that article has become much better and now I recommend this article for publication.